# Recombinant Art v4.01 protein produces immunological tolerance by subcutaneous immunotherapy in a wormwood pollen-driven allergic asthma female mouse model

Tao Wang[1,2], Jiaoni Chi[1], Zhimin Li[3], Yue Zhang[1], Yaojun Wang[3,4], Ming Ding[1], Bin Zhou[3], JiaChen Gui[1], Qiang Li[1]*

1 Department of Dermatology, Air Force Medical Center, PLA, Beijing, China, 2 Department of Dermatology, West China Hospital, Sichuan University, Chengdu, China, 3 Graduate School, Hebei North University, Zhangjiakou, China, 4 Handan Second Hospital, Hebei, China

☯ These authors contributed equally to this work.

* 16585260@qq.com

**Data Availability Statement:** All relevant data are within the article and its Supporting Information files.

## Abstract

Art v4.01 is a well-known profilin protein belonging to the pan-allergens group and is commonly involved in triggering allergic asthma, polyallergy, and cross-sensitization. It is also referred to as Wormwood due to its origin. Crude wormwood extracts are applied for allergen-specific immunotherapy (AIT). Whether the recombinant Art v4.01 (rArt v4.01) can produce *in vivo* immunological tolerance by subcutaneous immunotherapy (SCIT) remains elusive. In this study, to investigate the *in vivo* immunological response of rArt v4.01, Th2, Th1, Treg, Th17 type-related cytokines and phenotypes of immune cells were tested, facilitating the exploration of the underlying mechanisms. The expression and purification of Art v4.01 were carried out using recombinant techniques. Allergic asthma female BALB/c mice were induced by subcutaneous sensitization of wormwood pollen extract and intranasal challenges. SCIT without adjuvant was performed using the rArt v4.01 and wormwood pollen extract for 2 weeks. Following exposure to challenges, the levels of immunoglobulin E (IgE), cytokines, and inflammatory cells were assessed through enzyme-linked immunosorbent assay (ELISA) and histological examination of sera, bronchoalveolar lavage fluid (BALF), and lung tissue. These parameters were subsequently compared between treatment groups receiving rArt v4.01 and wormwood pollen extract. The rArt v4.01 protein was expressed, which had a high purity (>90%) and an allergenic potency. Compared with the pollen extract, rArt v4.01 was superior in terms of reducing the number of white blood cells (WBCs), total nucleated cells (TNCs), and monocytes (MNs) in BALF and the degree of lung inflammation (1.77±0.99 *vs.* 2.31±0.80, $P > 0.05$). Compared with the model group, only rArt v4.01 reduced serum IgE level (1.19±0.25 *vs.* 1.61±0.17 μg/ml, $P = 0.062$), as well as the levels of Th2 type-related cytokines (interleukin-4 (IL-4) (107.18±16.17 *vs.* 132.47±20.85 pg/ml, $P < 0.05$) and IL-2 (19.52±1.19 *vs.* 24.02±2.14 pg/ml, $P < 0.05$)). The study suggested that rArt v4.01 was superior to pollen extract in reducing the number of inflammatory cells in BALF, pneumonitis, levels of pro-inflammatory cytokines, and serum IgE level.

**Funding:** This study was funded by a Fund program: Natural Science Foundation of Beijing Municipality (7222186), Military Logistics Research Fund (BJ21J015, BKJ16J007, 2021ZY024). In this study, the funders play a major role in study design, decision to publish, and revision of the manuscript.

**Competing interests:** The authors have declared that no competing interests exist.

These findings confirmed that Art v4.01 could be a potential candidate protein for allergen-specific immunotherapy.

## Introduction

Allergic asthma is a chronic respiratory inflammatory disease caused by allergens, which is characterized by elevated immunoglobulin E (IgE) level, and its incidence is gradually increasing due to industrialization and the increase of inhaled allergens [1, 2]. Allergen-specific immunotherapy (AIT) is a disease-modifying, antigen-specific, and long-lasting therapy for allergic diseases [3]. Conventional AIT is mainly combined with whole allergen extracts, while natural extracts have several disadvantages, including variable composition and allergen content, as well as contamination with other allergenic sources [4]. These drawbacks can be overcome by recombinant allergens, which can be produced with a consistent quality and preserve IgE reactivity and T-cell epitopes [5]. The vaccine can be formulated according to the patient's needs, and the formulations contain only the relevant allergenic components [6]. Furthermore, it has been shown that immunotherapy with individual peanut components could induce protective effects against challenge with the whole peanut allergen through bystander effect [7]. Thus, it was confirmed that a single recombinant allergen could be applied in the study of AIT.

Profilin is a family of highly conserved proteins present in all eukaryotic cells that can regulate the activity of the microfilament system and intracellular calcium level [8]. It not only acts as a food allergen, mainly inducing oral allergy syndrome, but also it can act as an aeroallergen in a significant proportion of profilin-sensitized patients [9, 10]. Moreover, Wopfner et al. found that mugwort profilin strongly inhibited IgE-binding to profilin in extracts from birch pollen, celery, carrot, soy, and peanut, demonstrating extensive IgE cross-reactivity of recombinant mugwort profilin and profilin from pollen and plant-derived foods [11]. In addition, Muehlmeier et al. indicated that the poly-allergenicity of pollen was mainly due to the presence of profilin protein and calcium binding protein, of which 82% of patients with polyallergic symptoms were allergic to profilin [12]. Meanwhile, Valenta et al. confirmed that the purified profilin could significantly induce histamine release in profilin-allergic patients, as well as its common IgE-binding capacity, thereby characterizing profilin as a functional panallergen [13]. In a previous study, it was found that among individuals suffering from pollen allergy, the sensitization rate of profilin differed across various plants, with Bet v2 showing 21%, Hel a2 exhibiting 30.%, and palm showing a range of 12.3–30% [8, 14]. Therefore, given its important role in allergic diseases, scholars applied profilin to AIT. Nucera et al. found that desensitization of profilin protein reduced specific IgE level for food allergic patients, and decreased the mean diameter of wind clumps in skin prick tests. In addition, after 10 months of treatment, 7 patients exhibited increased tolerance to multiple foods that they were previously unable to tolerate [8]. It suggests that profilin produced positive effects in terms of clinical efficacy and immune response to AIT.

Wormwood cDNA clones coding for two isoforms of the panallergen profilin, wormwood profilin-1 and profilin-2(Art v4.01, Art v4.02), were first isolated and characterized by Nicole Wopfner in 2002, and no significant differences in the IgE binding properties of the two profilin isoforms were found [11, 13]. Thus, the present research was carried out to investigate Art v4.01, and it was found that 36% of individuals who suffered from wormwood pollen allergy showed positive results for IgE antibodies against both natural and recombinant wormwood profilin [15]. Despite not being a highly sensitizing protein, profilin plays a crucial role in individuals with polyallergies and cross-sensitization between pollen and plant-based foods. Due

to this, in the present study, an experiment was designed to administer subcutaneous immuno-therapy (SCIT) on allergic asthmatic mice induced by wormwood, specifically targeting Art v4.01.

In the present study, a mouse model of wormwood-induced allergic asthma was established, and the recombinant wormwood profilin (rArt v4.01) was utilized for SCIT without an adjuvant of allergic mice to assess different immunological characteristics and clinical effects. The results of this study may provide a theoretical basis for the optimization of SCIT.

## Materials and methods

### Vector construction and transformation of Escherichia coli

A harboring vector for expression in *E. coli* containing Art v4.01 cDNA was synthesized commercially (pET-28a; KMD Bioscience, Tianjin, China). Briefly, the Art v4.01 gene (GenBank Accession No. AJ421030) was designed with a 6xHis tag at the C-terminus, and the Art v4.01 gene was inserted between the BlpI and XbaI sites of the pET-28a vector (Fig 1A). The pET-28a-Art v4.01 vector plasmid was then transformed into BL21 (DE3) *E. coli* (KMD Bioscience) via heat shock for 90 s, incubated on ice quickly, and plated on Luria-Bertani (LB) medium containing ampicillin (Fig 1A and 1B).

### Expression, identification, purification, electrophoresis, and immunoblotting of Art v4.01 protein

The pET-28a-Art v4.01 vector fusion protein was expressed in BL21 (DE3) *E. coli* and induced by isopropyl-beta-thiogalactopyranoside (IPTG) according to the manufacturer's instructions. Theprotein samples were identified by 12% sodium dodecyl-sulfate polyacrylamide gel electrophoresis (SDS-PAGE), and they were stained using Coomassie brilliant blue. The proteins were purified by Ni2+ affinity chromatography and eluted, including resultant protein that was dialyzed and lyophilized. Finally, potential endotoxins in the rAlt v4.01 protein were

**A**

MSWQTYVDDHLMCDIEGTGQHLTSAAIFGTDGTVWAKSASFPEFKPNEIDAIIKEFNEAGQLAPTGLFLG GAKYMVIQGEAGAVIRGKKG

AGGICIKKTGQAMVFGIYDEPVAPGQCNMVVERLGDYLLDQGM HHHHHH

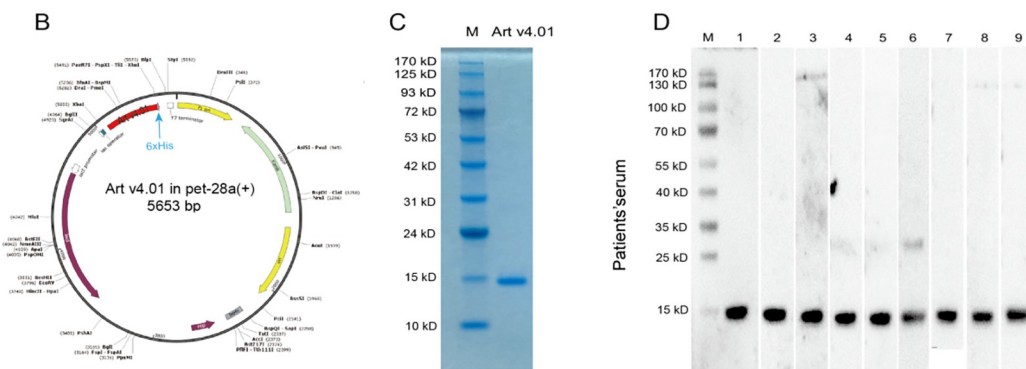

**Fig 1. A The amino acid sequence of Art v4.01**, and 6xHis tag is marked in blue. **B Construction of Escherichia coli protein expression vector for Art v4.01.** Art v4.01 gene has a 6xHis tag at the C-terminus. Gene is marked in red and His tag is marked in blue. **C SDS-PAGE analysis of rArt v4.01. D Western blotting of rArt v4.01 with serum from patients.** M: Molecular weight marker.

removed by endotoxin removal columns, and the lipopolysaccharide (LPS) level was detected at a concentration of < 10 EU/mg (Genscript Toxinsensor kit, L00350; Genscript Biotech Corp., Piscataway, NJ, USA).

Purity of Art v4.01 was identified by SDS-PAGE. For immunoblotting, proteins were transferred to nitrocellulose membranes, which were blocked with 3% bovine serum albumin (BSA) overnight, and incubated in diluted (1:5) serums from 9 allergic patients who resided in the prairie region of Hohhot, Inner Mongolia, China, and whose *Artemisi*a-specific IgE (sIgE) level was > 50 IU/ml for 2 h at 37˚C. Biotin-labeled goat anti-human IgE was diluted with Tris-buffered saline with Tween-20 (TBST), and it was added to the membrane and incubated for 2 h at 37˚C. Horseradish peroxidase (HRP)-labeled streptavidin was then diluted with TBST and incubated at 37˚C for 1.5 h. The membranes were thrice washed after each operation. The results were analyzed by enzyme-linked chemiluminescence. Finally, the concentration of rArt v4.01 was 0.4 mg/ml, as detected using the BCA Protein Assay kit (Solarbio Science & Technology Co., Ltd., Beijing, China).

In addition, B- and T-cell epitopes were predicted using immunoinformatic tools on the Art v4.01, including Bioinformatics Predicted Antigenic Peptides (BPAP), ABCpred, Immune Epitope Database Analysis Resource (IEDB-AR), NetMHC 4.0, SYFPEITHI, IEDB, NetMH-CII-2.3, and NetMHC II pan-3.2. The three-dimensional (3D) model of Art v4.01 (PDB code 6B6J) used in this study was obtained with the previously published structures as a primary model [16, 17], in which molecular graphics of 3D structures and surfaces were rendered using PYMOL [17].

## Wormwood pollen extracts

Briefly, wormwood pollen was crushed in liquid nitrogen, defatted with acetone at 4˚C for 8 h, protein was extracted by stirring with ammonium bicarbonate for 24 h, filtered, removed acetone, dialyzed against distilled water for 24 h, aliquoted, and lyophilized [18]. Then, the BCA method was employed to measure the extracted pollen protein concentration, and the protein samples were analyzed by SDS-PAGE. The LPS level was detected at a concentration of >10 EU/mg (Genscript Toxinsensor kit, L00350).

## Animals

Female BALB/c mice (specific pathogen-free grade; age, 5-6-week-old; body weight, 16–20 g) were purchased from the Animal Center of Shanghai Laboratory, Chinese Academy of Sciences (Shanghai, China), which fed in a specific pathogen-free grade breeding room. All animal experiments were approved by the Institutional Animal Ethics Committee (Approval No. QD-20210405001).

## Mouse sensitization

The animal sensitization model and specific immunotherapy were designed (Fig 2A), as previously described [5, 18, 19]. BALB/c mice were randomly divided into four groups (n = 8), including control group, model group, rArt v4.01 SCIT group (rArt v4), and wormwood pollen extract SCIT group (Extract). BALB/c mice were sensitized via subcutaneous injection on days 1, 8, 15, and 22 with 50 μg of pollen extract adsorbed to 2 mg of alum (77161; Thermo Fisher Scientific, Waltham, MA, USA) in 200 μl phosphate-buffered saline (PBS). On days 29, 30, and 31, challenges were performed by intranasal inoculation of 200 μg pollen extract in 20 μl PBS under 4.5% isoflurane anesthesia. Equal volumes of PBS were administered for only the control group at all time points.

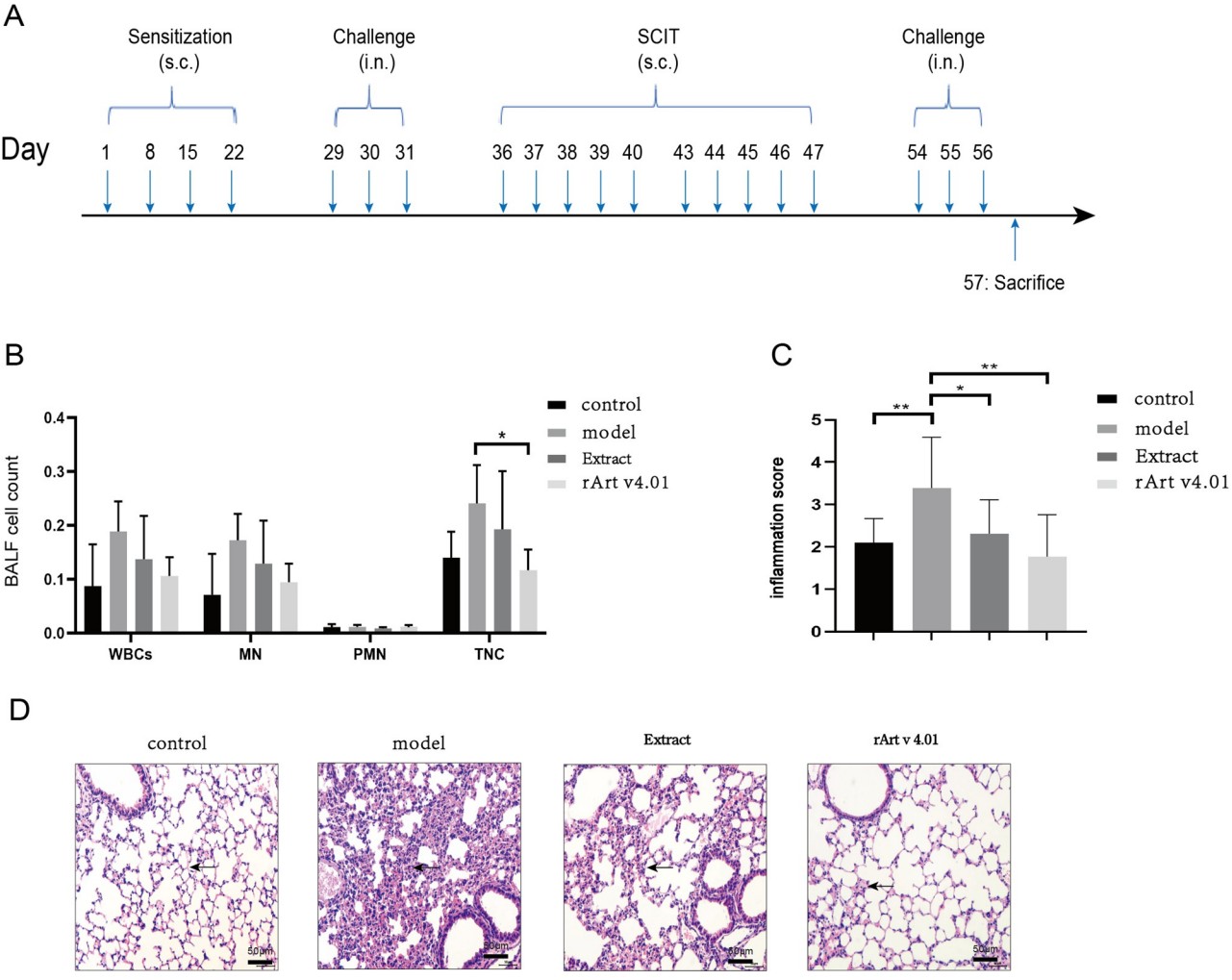

**Fig 2. A An experimental design.** The sensitization scheme of mice by wormwood pollen extract and their desensitization with vaccine based on recombinant Art v4.01 and pollen extract, as well as extract challenges, is depicted. SCIT: subcutaneous immunotherapy. sc, subcutaneous injection; in, nasal inhalation. In addition, RArt v4.01 suppresses lung inflammation in wormwood-induced asthmatic mice. **B Comparison of BALF cell counts between SCIT treatment groups. C The inflammation scores in lung histology.** The scores were analyzed by one-way analysis of variance with Tukey's multiple comparison test. **D H&E staining of the lung tissues.** Scale bar = 50 μm. Arrows point to the inflammatory cell infiltrate. Data represent the mean ± SD. *, $P < 0.05$; **, $P < 0.01$; ****, $P < 0.0001$. Replicates of these experiments are shown in S2 Fig.

## SCIT

SCIT was carried out with 50 μg rArt v4.01 or pollen extract in 200 μl PBS on days 36–40 and 43–47. Mice in the control and model groups received equal volumes of PBS. On days 54, 55, and 56, challenges were performed by intranasal inoculation of 200 μg pollen extract in 20 μl PBS under isoflurane anesthesia. All mice were sacrificed via cervical dislocation on day 57, and samples were collected for further analysis (Table 1).

## Determination of IgE, IgG1, and IgG2a levels

Blood was collected by cardiac puncture under 4.5% isoflurane anesthesia, and the total serum levels of IgE, IgG1, and IgG2a were measured using the mouse IgE (70-EK275-96), IgG1 (70-EK271-96), and IgG2a (70-EK273-96) enzyme-linked immunosorbent assay (ELISA) kits (BioMedBAY Co., Ltd., Shanghai, China), according to the manufacturer's instructions.

**Table 1. Outline of the SCIT protocol (n = 8/group).**

| Group | Sensitization (sc) | Challenge (in) | SCIT (sc) | Challenge (in) |
|---|---|---|---|---|
| control | 200μl, PBS | 20μl, PBS | 200μl, PBS | 20μl, PBS |
| model | 200μl, 50μg extract /2mg Alum | 20μl, 200μg extract | 200μl, PBS | 20μl, 200μg extract |
| rArt v4.01 | 200μl, 50μg extract/2mg Alum | 20μl, 200μg extract | 200μl, 50μg rArt v4.01 | 20μl, 200μg extract |
| Extract | 200μl, 50μg extract /2mg Alum | 20μl, 200μg extract | 200μl, 50μg extract | 20μl, 200μg extract |

Art v, Artemisia vulgaris (wormwood); Alum, Aluminum; SCIT, subcutaneous immunotherapy; sc, subcutaneous injection; in, nasal inhalation.

## Assessment of cellular immune response by cytokine profile

The removed spleen was rapidly homogenized in 5 ml of radioimmunoprecipitation assay (RIPA) buffer. Then, it was centrifuged at 12,000 g for 20 min at 4°C, and the supernatant was stored at -80°C.

The supernatant was used for cytokine measurement by indirect ELISA, following the manufacturer's instructions (BioMedBAY Co., Ltd.), including interleukin-4 (IL-4) (70-EK204/2-48), IL-10 (70-EK210/4-48), interferon gamma (IFN-γ) (70-EK280/3-48), transforming growth factor-β (TGF-β) (70-EK981-48), tumor necrosis factor-α (TNF-α) (70-EK282HS-48), IL-2 (70-EK202/2-48), IL-17A (70-EK217HS-48), and IL-5 (70-EK205-96).

## Analysis of the BALF

When mice were euthanized, the lavage of lung tissue was immediately performed by the trachea cannula with 1 ml of PBS, and the BALF was slowly collected after 5 bouts of pumping, with a recovery rate of >80% (0.8 ml). The precipitate was dissolved in 100 μl of PBS and resuspended, and the number of white blood cells (WBCs), monocytes (MNs), polymorphonuclear neutrophils (PMNs), and total nucleated cells (TNCs) was counted by a hemocytometer (XN-1000V; Sysmex Co., Ltd., Tokyo, Japan).

## Histological analysis of murine lungs

Lung tissues were preserved overnight in 4% paraformaldehyde for histological analysis. Paraffin-embedded lung tissues (thickness, 5 mm) were subjected to hematoxylin and eosin (H&E) staining to observe lung tissue morphology, airway epithelial damage, airways, and perivascular eosinophils (Eos) under a randomized field of view of 40% objective using a light microscope, and scored according to the Underwood criteria [20].

## Statistical analysis

Data were analyzed using GraphPad Prism 8.0 software (GraphPad Software Inc., San Diego, CA, USA), and data were expressed as mean ± standard deviation (X ± SD). Each group of data was first tested for normality, and normally distributed data were analyzed by the Chi-square test. Abnormally distributed data were analyzed by one-way analysis of variance (one-way ANOVA), and differences were considered statistically significant at $P < 0.05$.

In order to ensure rigor and reproducibility of this experiment, we repeated the most important readouts to ensure that the therapy works consistently, and the data of the repeated experiment were shown in the supplement material.

## Results

### The expressed rArt v4.01 exhibited a high purity and an allergenic potency

The allergen used in this study was Art v4.01, a subtype of Art v4. We produced Art v4.01 consisted of 139 amino acids with a theoretical molecular weight of 15.0 kDa. SDS-PAGE assay showed that the profilin protein of wormwood pollen had a clear band at a relatively molecular mass (*Mr*) of 15,000 and a high purity (>90%) (Fig 1C).

Then, to verify the allergenic potency of Art v4.01, Western blotting was performed using sera from 9 *Artemisia* pollen-allergic patients. The results showed that Art v4.01 could bind to IgE epitopes of allergic samples (Fig 1D).

Finally, among 5 B-cell epitopes at 14–20, 39–44, 60–63, 91–97, and 109–115 amino acid sites, one CD4+ T-cell epitope at 71–85 aa and one CD8+ T-cell epitope at 119–127 aa were successfully predicted (S1 Fig, S1 and S2 Tables).

### Wormwood-induced allergic asthma mouse model

Compared with the control group, the number of WBCs, TNCs, PMNs, and MNs showed an upward trend in BALF of the model group (P > 0.05, Fig 2B). Examination of lung tissue histopathology showed a rise in the population of mice with moderate alveolar wall congestion in the model group versus that in the control group. In addition, the presence of moderate inflammatory cell infiltration (MNs and PMNs), with predominantly monocyte infiltration, as well as the presence of mild/moderate emphysema, were manifested by an enlarged alveolar lumen and thin wall. The inflammation score was significantly higher in the model group than that in the control group (P < 0.01, Fig 2C). The results showed that mice in the model group produced a significant inflammatory response.

The serum IgE level of mice in the model group was 1.61±0.177 µg/ml, which was significantly higher than that in the control group (0.93±0.41 µg/ml, P < 0.05, Fig 3A), while serum IgG1 and IgG2a levels did not significantly change between the two groups. In addition, the levels of T helper cell (Th)-2 cytokines (IL-4 and IL-5) in the spleen tissue of mice in the model group were higher than those in the control group (P > 0.05, Fig 3B); the levels of T helper cell (Th)-1 cytokines (IFN-γ) were lower than those in the control group (P > 0.05, Fig 3B), and the levels of Treg-related cytokines (IL-10) were significantly lower than those in the control group (P < 0.05, Fig 3B).

### Effects and mechanisms of wormwood and rArt v4.01 on murine allergic asthma

In order to compare the immunological and clinical effects between rArt v4.01 and wormwood pollen extract desensitization, asthmatic mice were treated with rArt v4.01, wormwood pollen extract, and PBS.

Compared with the model group, the number of WBCs, TNCs, MNs, and PMNs in the BALF in the treatment group showed a downward trend, in which the number of TNCs in the rArt v4.01 group significantly decreased (P < 0.05, Fig 2B), and the number of inflammatory cells in the extract group did not significantly vary (P > 0.05). In the repeated experiment, the number of inflammatory cells in the BALF in the treatment group was similarly decreased, in which the number of WBCs, MNs in the rArt v4.01 group significantly decreased (P < 0.05, S2A Fig), whereas there was no significant vary in the extract group(P > 0.05, S2A Fig). The inflammation scores in the rArt v4.01 and extract groups [1.77±0.99 and 2.31±0.80, respectively] were significantly lower than those in the model group (3.39±1.2, P < 0.05, Fig 2C). Similarly, in the repeated experiment, the inflammation scores in the rArt v4.01 and extract

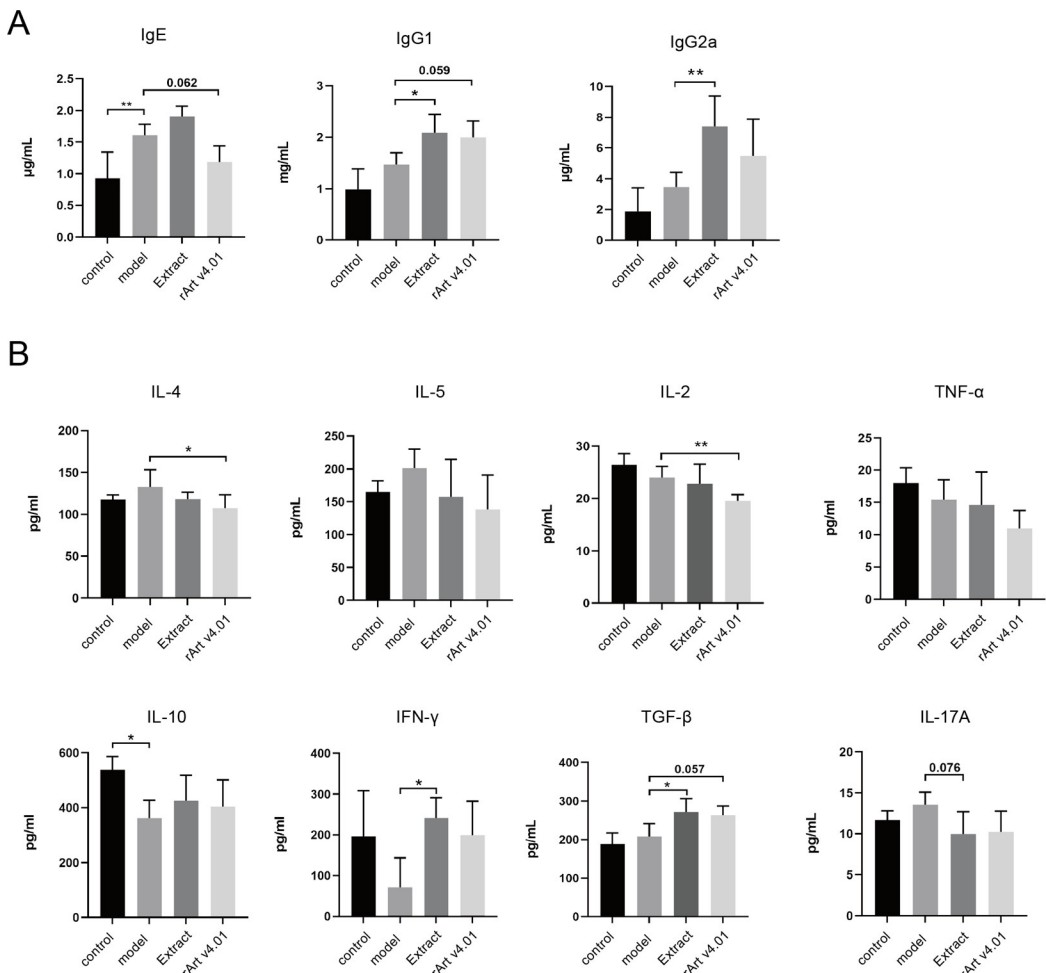

**Fig 3. A Quantification of total serum IgE, IgG1, and IgG2a levels** in short-term treatment groups of BALB/c mice. **B Quantification of the levels of cytokines**, such as IL-4, IL-5, IL-2, IL-17A, IL-10, TNF-α, IFN-γ, and TGF-β were analyzed by ELISA in spleen homogenate in short-term treatment groups of BALB/c mice. Data represent the mean ± SD. *, $P < 0.05$; **, $P < 0.01$. Replicates of these experiments are shown in S3 Fig.

groups [2.30±0.64 and 2.95±0.61, respectively] were significantly lower than those in the model group (4.20±1.19, $P < 0.05$, S2B Fig). The results indicated that rArt v4.01 was superior to extract in suppressing lung inflammation in mice.

In addition, serum IgE levels in the rArt v4.01 and extract groups were 1.19±0.25 and 1.91 ±0.16 µg/ml, respectively, in which serum IgE level in the rArt v4.01 group was lower than that in the model group (1.61±0.17) µg/ml ($P = 0.062$), while the extract group had no significant effect ($P > 0.05$, Fig 3A). In contrast, serum IgG1 levels in the rArt v4.01 and extract groups were 1.99±0.32 and 2.08±0.38 mg/ml, respectively, which were significantly higher than those in the model group (1.47±0.23 mg/ml, $P = 0.059$, $P < 0.05$, Fig 3A). Similar to IgG1, serum IgG2a level was upregulated in the SCIT group. Importantly, serum IgG2a level was significantly higher in the extract group than that in the model group (7.37±2.00 *vs*. 3.46±0.96 µg/ ml) ($P < 0.05$), while rArt v4.01 had no significant effect ($P > 0.05$, Fig 3A). Similarly, in the repeated experiment, serum IgE level in the rArt v4.01 group was lower than that in the model group (1.17±0.18) µg/ml ($P = 0.051$), while the extract group had no significant effect ($P > 0.05$, S3A Fig). Serum IgG1 levels in the extract groups were 1.97±0.19 mg/ml, which

were significantly higher than those in the model group (1.56±0.03 mg/ml, P < 0.05, S3A Fig). And serum IgG2a level was significantly higher in the extract group than that in the model group (6.44±1.55 vs. 3.68±0.68 μg/ml) (P < 0.05), while rArt v4.01 had no significant effect (P > 0.05, S3A Fig).

The levels of Th2-related cytokines (IL-4 and IL-2) were significantly downregulated in the rArt v4.01 group compared with those in the model group (P < 0.05, Fig 3B). In contrast, the levels of Treg-related cytokine (TGF-β) and Th1-type cytokine (IFN-γ) were significantly upregulated in the extract group (P < 0.05). In addition, the levels of IL-17A, IL-5, and TNF-α did not significantly differ between the two treatment groups (P > 0.05, Fig 3B). In the repeated experiment, the levels of Th2-related cytokines (IL-5 and IL-2) and Th17-related cytokines(IL-17A) were significantly downregulated in the rArt v4.01 and extract group compared with those in the model group (P < 0.05, S3B Fig), with a greater decrease in the rArt v4.01 group compared to the extract group. TNF-α was significantly downregulated in the rArt v4.01 group compared with the model group (P < 0.05, S3B Fig). In contrast, the levels of Treg-related cytokine (IL-10) and Th1-type cytokine (IFN-γ) were significantly upregulated in the extract group (P < 0.05), TGF-β significantly upregulated in the rArt v4.01 and extract group compared with those in the model group (P < 0.05, S3B Fig). In addition, the levels of IL-4 did not significantly differ between the two treatment groups (P > 0.05, S3B Fig).

## Discussion

To date, several studies have concentrated on AIT for wormwood, as the main airborne pollen of wormwood spp. in the world. However, the majority of them mainly concentrated on the wormwood pollen extracts [21, 22], and only wormwood has been studied for a hypoallergenic vaccine of Art v1 [23, 24], as well as a T-cell epitope fusion vaccine of Art v1 with ragweed Amb a1 [25, 26]. Thus, there is a lack of exploration of Art v4.

RArt v4.01 possesses the advantages of high-quality vaccines and epitope ratios, accompanying by retention of IgE reactivity and induction of allergen-specific blocking IgG antibody production [26]. More importantly, rArt v4.01 excludes non-protein components, such as chitin, β-glucan, and endotoxin contained in the crude extract, interfering with the induction of immune tolerance by activating a pro-inflammatory innate immune response [27]. Furthermore, a previous study has demonstrated that recombinant allergens are as efficacious as allergen extracts and purified allergens, and they have the same side effects as natural allergens [5]. In the present study, a mouse model of allergic asthma induced by wormwood pollen extract (wormwood) was established. The number of inflammatory cells and serum IgE level in BALF of allergic asthmatic mice were significantly risen, which was similar to the results of previous studies [18, 28, 29]. The lung tissues of mice showed signs of allergic reactions, such as congestion and inflammatory cell infiltration, and inflammation scores were significantly elevated. It indicated that the mouse model of wormwood-induced allergy was successfully established.

In the present study, mice with allergic asthma were treated with recombinant profilin protein (rArt v4.01) and wormwood pollen extract. It was demonstrated that both strategies mitigated allergic inflammation, including a decrease in the number of inflammatory cells in BALF, and reduced congestion and inflammatory cell infiltration in the alveolar wall. However, the ability of rArt v4.01 to inhibit inflammation was superior to that of wormwood, which could be attributed to the pro-inflammatory effect of impurities on the crude materials interfering with the therapeutic effect of AIT [27].

In the present study, it was also confirmed that rArt v4.01 inhibited Th2 response more strongly than crude extract, while induced Th1, regulatory T cells (Tregs) weaker than crude extract. A marker of successful AIT is the correction of the imbalance between Th1 and Th2

responses [30]. In the first experiment, rArt v4.01 was dominated by inhibition of the levels of Th2 cell-associated factors (IL-4 and IL-2), in which IL-2 could induce conversion of Th0 to Th2 cells [31]. It indicated that rArt v4.01 could inhibit Th2 response more than wormwood, which is similar to Hesse et al.'s finding [27]. However, in the second experiment, rArt v4.01 was dominated by inhibition of the levels of IL-17A and TNF-α, which was previously described to control the recruitment of eosinophils, neutrophils, and T cells into the lung [32]. It indicated that rArt v4.01 could inhibit the production of pro-inflammatory cytokines more than wormwood in the two experiments. These results may be related to the higher protein content in the recombinant protein vaccine than in the crude extract [27]. In addition, wormwood was dominated by the induction of Th1-type cytokine (IFN-γ), IL-10 and TGF-β production in the two experiments, of which TGF-β is an inhibitory cytokine produced by Tregs, inducing apoptosis and incompetence of effector T cells [28, 33], and IL-10 is produced by both Tregs and Bregs, is a potent suppressor of both total and allergen-specific IgE while it simultaneously increases IgG4 production [34].

The antigen-specific neutralizing antibodies are considered as a biomarker for AIT [5]. However, it has been shown that the therapeutic effect of pollen allergy on mice has a greater correlation with total serum IgE level rather than Art v1-sIgE [29]. Heldner et al. demonstrated that low doses of dust mite allergens reduced T-IgE level, while did not affect sIgE level [32]. In the present study, only recombinant protein treatment significantly reduced total serum IgE level. Importantly, total serum IgE level was elevated in the wormwood group and serum IL-4 level was not significantly reduced, and this phenomenon could be attributed to the pro-inflammatory effect of the non-protein component of the crude extract [27]. In addition, this paradoxical change in IgE level could be correlated with the treatment method, drug concentration, and route of administration [35]. Another marker for successful AIT is the induction of sIgG4 production, while mouse IgG1 and IgG2a are considered as structural and functional homologs of human IgG4 [32]. In the present study, it was further confirmed that only the wormwood was effective in inducing serum levels of IgG1 and IgG2a, while rArt v4.01 did not produce similar results. Similarly, Mar´ıa et al. demonstrated that a robust synthesis of IgG2a was observed only when ISS was used as an adjuvant for AIT, and no significant response was observed with any adjuvants, such as Alum, MPL, CalPh, and Sal [36]. Hence, it can be confirmed that adjuvant is very important for the production of IgG. Therefore, the failure of IgG production in this study was due to the adjuvant-free subcutaneous immunotherapy and the fact that the protein was a minor allergen, which might require AIT in combination with other allergens to induce more significant immune tolerance. In addition, another reason justifying the failure of IgG production was that endotoxin contamination in the rArt v4.01 preparations used in this study was low, thus, the immune system was not triggered towards Th1 upon immunotherapy [37].

The present study has also some limitations. Firstly, this study lacks the assessment of long-term efficacy after treatment to determine the duration of the protective effect after SCIT. Secondly, the predicted epitopes were not experimentally validated. Thirdly, this study did not assess the allergen-specific Ig levels. Although total antibody levels can reflect the ability of allergens to induce immune tolerance, allergen-specific Ig levels can refine the assessment of the efficacy for AIT. So later experiments will be supplemented with tests of these biomarkers. Finally, only female mice were used in this study, and it was not possible to exclude the influence of gender effect on the results of the study. In addition, some studies have shown that airway responsiveness to methacholine was higher in male mice [38, 39]. So gender effect should be further investigated in the future studies.

In conclusion, this study showed that both rArt v4.01 and pollen extract suppressed inflammation in allergic asthmatic mice. RArt v4.01 was superior to pollen extract, reducing the

number of inflammatory cells in BALF, pneumonitis, levels of pro-inflammatory cytokines, and serum IgE level. In contrast, pollen extract was superior to rArt v4.01 in the induction of levels of Th1, Treg-related cytokines, and IgG1 and IgG2a. The findings may contribute to further understanding of the mechanisms of wormwood sensitization and SCIT. Moreover, the data confirmed that Art v4.01 could be a potential candidate protein for allergen-specific immunotherapy in polyallergic patients and cross-sensitization between pollen and plant-derived food.

## Supporting information

**S1 Fig. Surface representation of predicted B- and T-cell epitopes of Art v4.01.** A Epitopes of Art v4.01 were highlighted on the amino acid sequence. B, C: B-cell and T-cell epitope positions identified on the surface of the 3D structure of Art v4.01.
(DOCX)

**S2 Fig. These data are the result of repeated experiments, including BALF analysis and mouse lung histology.**
(DOCX)

**S3 Fig. These data are the result of repeated experiments, including antibody and cytokine analysis.**
(DOCX)

**S4 Fig. The original underlying images of SDS-PAGE analysis for rArt v4.01 in Fig 1C.**
(DOCX)

**S5 Fig. The original underlying images of Western blotting for rArt v4.01 with serum from patients in Fig 1D.**
(DOCX)

**S1 Table. Prediction results of B-cell epitopes of Art v4.01.**
(DOCX)

**S2 Table. Prediction results of CD4+ T and CD8+ T cell epitopes of Art v4.01.**
(DOCX)

**S1 Raw image.**
(PDF)

## Author Contributions

**Conceptualization:** Tao Wang, Qiang Li.

**Data curation:** Tao Wang, Jiaoni Chi, Zhimin Li, Yue Zhang, Bin Zhou, JiaChen Gui.

**Formal analysis:** Tao Wang, Jiaoni Chi, Zhimin Li, Yue Zhang, Ming Ding.

**Funding acquisition:** Qiang Li.

**Investigation:** Jiaoni Chi, Zhimin Li.

**Supervision:** Qiang Li.

**Validation:** Tao Wang.

**Visualization:** Tao Wang.

**Writing – original draft:** Tao Wang.

**Writing – review & editing:** Jiaoni Chi, Zhimin Li, Yue Zhang, Yaojun Wang, Qiang Li.

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
