## [Decision Letter · Decision Letter 0]

17 Apr 2023

PONE-D-23-00019Recombinant Art v4.01 Protein Produces Immunological Tolerance by Subcutaneous Immunotherapy in a Wormwood Pollen-driven Allergic Asthma Mouse ModelPLOS ONE

Dear Dr. Li,

Thank you for submitting your manuscript to PLOS ONE. After careful consideration, we feel that it has merit but does not fully meet PLOS ONE’s publication criteria as it currently stands. Therefore, we invite you to submit a revised version of the manuscript that addresses the points raised during the review process.

We look forward to receiving your revised manuscript.

Kind regards,

Shih-Chang Hsu

Academic Editor

PLOS ONE

Journal Requirements:

2. To comply with PLOS ONE submissions requirements, in your Methods section, please provide additional information regarding the experiments involving animals and ensure you have included details on (1) methods of sacrifice, (2) methods of anesthesia and/or analgesia, and (3) efforts to alleviate suffering."

"This study was funded by a Fund program: Natural Science Foundation of Beijing Municipality (7222186), Military Logistics Research Fund (BJ21J015, BKJ16J007, 2021ZY024)."               

4. PLOS requires an ORCID iD for the corresponding author in Editorial Manager on papers submitted after December 6th, 2016. Please ensure that you have an ORCID iD and that it is validated in Editorial Manager. To do this, go to ‘Update my Information’ (in the upper left-hand corner of the main menu), and click on the Fetch/Validate link next to the ORCID field. This will take you to the ORCID site and allow you to create a new iD or authenticate a pre-existing iD in Editorial Manager. Please see the following video for instructions on linking an ORCID iD to your Editorial Manager account: https://www.youtube.com/watch?v=_xcclfuvtxQ.

Reviewers' comments:

Reviewer's Responses to Questions

**Comments to the Author**

1. Is the manuscript technically sound, and do the data support the conclusions?

Reviewer #1: Partly

Reviewer #2: Partly

2. Has the statistical analysis been performed appropriately and rigorously? 

Reviewer #1: Yes

Reviewer #2: Yes

3. Have the authors made all data underlying the findings in their manuscript fully available?

Reviewer #1: Yes

Reviewer #2: Yes

4. Is the manuscript presented in an intelligible fashion and written in standard English?

Reviewer #1: Yes

Reviewer #2: No

5. Review Comments to the Author

Reviewer #1: Dear Authors

Congratulation on your great work. The article appears scientifically sound, with an explicit method and outcome.

I got confused with tagged figures in the result and discussion. Namely, I couldn't find figures (C2, 3B, and 5B). Kindly review the supplement figures.

Reviewer #2: - The authors were trying to investigate the ability of the recombinant Art v4.01 to produce immunological tolerance in BALB/c mice model of allergic asthma in comparison to the use of crude wormwood extracts. However, the sample size (n=8) is not enough to achieve the main objective of this study.

- The manuscript needs extensive revision for language and grammar (Revision by a professional is recommended)

- In the materials and methods section, the groups of animals and different treatments should be identified more clearly. Furthermore, the whole section should be rephrased to avoid confusion.

- IgE binding capacity should be identified

- Many words are splitting at the end of lines

6. PLOS authors have the option to publish the peer review history of their article (what does this mean?). If published, this will include your full peer review and any attached files.

Reviewer #1: **Yes: **Siddig Eltyeb Yousif Abdallah

Reviewer #2: No

---

## [Author Response · Author response to Decision Letter 0]

29 May 2023

Dear Editor

Thank you for your letter and for the reviewers’ comments concerning our manuscript entitled “Recombinant Art v4.01 Protein Produces Immunological Tolerance by Subcutaneous Immunotherapy in a Wormwood Pollen-driven Allergic Asthma Mouse Model” . Those comments were greatly valuable and significant for revising and improving this manuscript. We have carefully studied the reviewers’ comments, and we have accordingly applied revisions. All the changes have been highlighted in red in the revised version of the manuscript, and we hope that the revised version of the manuscript would be acceptable for publication in your prestigious journal. The reviewers’ comments have been replied one-by-one as follows:

# Journal Requirements

1.Please ensure that your manuscript meets PLOS ONE's style requirements, including those for file naming.

Response: The manuscript was modified according to the PLOS ONE’s formatting rules, and all the changes have been highlighted in red in the revised version of the manuscript.

2.To comply with PLOS ONE submissions requirements, in your Methods section, please provide additional information regarding the experiments involving animals and ensure you have included details on (1) methods of sacrifice, (2) methods of anesthesia and/or analgesia, and (3) efforts to alleviate suffering.

Response: We described the methods of sacrifice was cervical dislocation on line 163, the methods of anesthesia was used 4.5% isoflurane on line 148, and the methods of alleviating suffering was anesthetized mice on line 167. 

3.Please state what role the funders took in the study.

Response: The funders play a major role in study design, decision to publish, and revision of the manuscript. And this amended role of funder statement in cover letter.

4.PLOS requires an ORCID iD for the corresponding author in Editorial Manager on papers submitted after December 6th, 2016. 

Response: The corresponding author’s ORCID iD was already associated with his account of PLOS ONE.

5.PLOS ONE now requires that authors provide the original uncropped and unadjusted images underlying all blot or gel results reported in a submission’s figures or Supporting Information files. 

Response: We provided the original underlying images for all blot and gel data reported in supporting information files, and described the original underlying images were placed in the supporting information files in cover letter.

6.Please review your reference list to ensure that it is complete and correct. If you have cited papers that have been retracted, please include the rationale for doing so in the manuscript text, or remove these references and replace them with relevant current references.

Response: The references had been corrected which no papers that have been retracted.

# Reviewer 1

Congratulation on your great work. The article appears scientifically sound, with an explicit method and outcome.

1.I got confused with tagged figures in the result and discussion. Namely, I couldn't find figures (C2, 3B, and 5B). Kindly review the supplement figures.

Response: 

1)Fig 2C represents the inflammation scores in lung histology which was in the middle part of the second image on lines 222 and 247.

2)Fig. 3B represents the quantification of the levels of cytokines, IL-4, IL-5, IL-2, IL-17A , IL-10, TNF-α, IFN-γ and TGF-β analysed by ELISA in spleen homogenate in short-term treatment groups of BALB/c mice studieds which was in the bottom part of the 3rd image on lines 228-230, 259 and 262.

.

3)We were very sorry, Fig. 5B on line 244 was an author’s error and should be corrected to Fig. 2B, which represents the comparison of BALF cell counts between SCIT treatment groups.

4)Supplemental Fig. 2 was corrected to S1 Fig. and Supplemental Tab. 2，3 was corrected to S1 Table and S2 Table on line 212.

# Reviewer 2

1.The authors were trying to investigate the ability of the recombinant Art v4.01 to produce immunological tolerance in BALB/c mice model of allergic asthma in comparison to the use of crude wormwood extracts. However, the sample size (n=8) is not enough to achieve the main objective of this study.

Response: Thanks for your significant comment. I agree with your view which the sample size (n=8) was not enough to achieve the main objective of this study. Meanwhile, we determined the number of mice based on the available literature, and the number of mice in their study was 8, 7, 6, and 6[1-4]. Therefore, I think that the results of this study could provide support for the main objective of this study, however, if the main objective of this study was to be achieved, more mice need to be performed to get more convincing results.

[1] Hesse L, van Ieperen N, Habraken C, Petersen AH, Korn S, Smilda T, et al. Subcutaneous immunotherapy with purified Der p1 and 2 suppresses type 2 immunity in a murine asthma model. Allergy. 2018;73(4):862-874. 

[2] Heldner A, Alessandrini F, Russkamp D, Heine S, Schnautz B, Chaker A, et al. Immunological effects of adjuvanted low-dose allergoid allergen-specific immunotherapy in experimental murine house dust mite allergy. Allergy. 2022;77(3):907-919.

[3] Liu J, Yin J. Immunotherapy With Recombinant Alt a 1 Suppresses Allergic Asthma and Influences T Follicular Cells and Regulatory B Cells in Mice. Front Immunol. 2021;12:747730. 

[4] Tabynov K, Babayeva M, Nurpeisov T, Fomin G, Nurpeisov T, Saltabayeva U, et al. Evaluation of a Novel Adjuvanted Vaccine for Ultrashort Regimen Therapy of Artemisia Pollen-Induced Allergic Bronchial Asthma in a Mouse Model. Front Immunol. 2022;13:828690. 

2.The manuscript needs extensive revision for language and grammar (Revision by a professional is recommended).

Response: We had carefully revised our manuscript, which was also polished by a native English speaker from a language services company.

3.In the materials and methods section, the groups of animals and different treatments should be identified more clearly. Furthermore, the whole section should be rephrased to avoid confusion.

Response: In order to more clearly identify the groups of animals and different treatments, negative control group and NC group was corrected to control group on lines 144, 149, 161, 215, 218, 221, 225 and 228-230; Positive control group and PC group was corrected to model group on lines 33, 144,161, 216, 218, 221-222, 224, 227, 242, 246, 250, 253, 255, 259; rArt v4.01 SCIT group abbreviated as rArt v4.01; wormwood pollen extract SCIT group abbreviated as Extract. sensitized subcutaneously was corrected to sensitized via subcutaneous injection on line 144; intranasal instillation was corrected to intranasal inoculation on lines 148, 162; ‘Mice were subcutaneously injected’ was corrected to ‘SCIT was carried out’ on line 160.

4.IgE binding capacity should be identified

Response:

IgE binding capacity of rArt v 4 to wormwood pollen specific IgE was examined using sera of the model group of mice based on Western blot and enzyme-linked immunosorbent reaction (ELISA). 

First of all, Western blot was performed using sera from wormwood pollen-allergic mice. The results showed that rArt v 4 was able to bind to IgE epitopes of allergic samples (Fig 1). It showed IgE reactivity in the serum from wormwood pollen-allergic mice. 

Fig 1. IgE binding to wormwood pollen extracts was investigated by immunoblot. Band around 35kD was found to exist as homodimers of recombinant Art v 4.

In addition, Indirect-ELISA was performed to determine sIgE against rArt v 4 using sera from wormwood pollen-allergic mice. Plates were coated with rArt v 4 (5ug/ml) in ELISA Binding Buffer (ScyTek laboratories, USA) and kept at 4℃ overnight. After washing three times with phosphate-buffered saline, plates were blocked with Super Block (ScyTek laboratories, USA) for 10 minutes. Then, plates were incubated for 2 hours with sera from wormwood pollen-allergic mice and blank control mice at 1:0, 1:1, 1:2, 1:4, 1:8, 1:16, 1:32, 1:64, 1:128 and 1:256 dilution, respectively. After washing, the goat anti-mouse IgE-HRP at 1:2000 dilution was added to the wells. The color was developed using TMB Soluble Reagent (ScyTek laboratories, USA) , and then absorbance at 450nm was measured. The results showed that sera from wormwood pollen-allergic mice could bind with rArt v 4 and (Fig 2) and the binding of sIgE to rArt v 4 is weakened as the sera dilution increase.

Fig 2. IgE binding to wormwood pollen extracts was investigated by ELISA

5.Many words are splitting at the end of lines

Response: The formatting of manuscript had been changed which words could not be split at the end of lines.

---

## [Decision Letter · Decision Letter 1]

25 Jul 2023

PONE-D-23-00019R1Recombinant Art v4.01 Protein Produces Immunological Tolerance by Subcutaneous Immunotherapy in a Wormwood Pollen-driven Allergic Asthma Mouse ModelPLOS ONE

Dear Dr. Li,

Thank you for submitting your manuscript to PLOS ONE. After careful consideration, we feel that it has merit but does not fully meet PLOS ONE’s publication criteria as it currently stands. Therefore, we invite you to submit a revised version of the manuscript that addresses the points raised during the review process.

Unfortunately, the required standards of the scientific rigor and reproducibility are still not met in the revised manuscript. The number of animals per group used is very small and the experiment was not independently repeated in a new cohort of animals to validate the results.

There is also an issue of the selective female mouse usage. When study is performed in one sex of animals, this has to be noted in the title and abstract, and the discussion need to address this limitation stating that the results in males could be different and should be further investigated in the future studies.

We look forward to receiving your revised manuscript.

Kind regards,

Michal A Olszewski, DVM, PhD

Academic Editor

PLOS ONE

Reviewers' comments:

Reviewer's Responses to Questions

**Comments to the Author**

1. If the authors have adequately addressed your comments raised in a previous round of review and you feel that this manuscript is now acceptable for publication, you may indicate that here to bypass the “Comments to the Author” section, enter your conflict of interest statement in the “Confidential to Editor” section, and submit your "Accept" recommendation.

Reviewer #1: All comments have been addressed

Reviewer #2: (No Response)

2. Is the manuscript technically sound, and do the data support the conclusions?

Reviewer #1: Yes

Reviewer #2: Partly

3. Has the statistical analysis been performed appropriately and rigorously? 

Reviewer #1: Yes

Reviewer #2: Yes

4. Have the authors made all data underlying the findings in their manuscript fully available?

Reviewer #1: Yes

Reviewer #2: Yes

5. Is the manuscript presented in an intelligible fashion and written in standard English?

Reviewer #1: Yes

Reviewer #2: Yes

6. Review Comments to the Author

Reviewer #1: dear Authors. congratulation for your work. You get respond to the comment and appeal my confusion

wish you all the best

Reviewer #2: The authors have poorly responded to some of the comments including the number of mice subjects which is one of the most important comments. Furthermore, the authors should justify their selection of female over male mice otherwise the experiment should use both male and female mice to exclude the gender effect. There should also be replication of the animal experiment to ensure fidelity.

7. PLOS authors have the option to publish the peer review history of their article (what does this mean?). If published, this will include your full peer review and any attached files.

Reviewer #1: No

Reviewer #2: No

---

## [Author Response · Author response to Decision Letter 1]

14 Sep 2023

Dear Editor

Thank you for your letter and for the reviewers’ comments concerning our manuscript entitled “Recombinant Art v4.01 Protein Produces Immunological Tolerance by Subcutaneous Immunotherapy in a Wormwood Pollen-driven Allergic Asthma Mouse Model” . Those comments were greatly valuable and significant for revising and improving this manuscript. We have carefully studied the reviewers’ comments, and we have accordingly applied revisions. All the changes have been highlighted in red in the revised version of the manuscript, and we hope that the revised version of the manuscript would be acceptable for publication in your prestigious journal. The reviewers’ comments have been replied one-by-one as follows:

# Journal Requirements

1.The number of animals per group used is very small and the experiment was not independently repeated in a new cohort of animals to validate the results. There is also an issue of the selective female mouse usage. When study is performed in one sex of animals, this has to be noted in the title and abstract, and the discussion need to address this limitation stating that the results in males could be different and should be further investigated in the future studies.

Response: These questions will be responded to in detail in the response to reviewers.

# Reviewer 1

1.Dear Authors. congratulation for your work. You get respond to the comment and appeal my confusion.

wish you all the best

Response: Dear reviewer, thank you very much for your comments , which were greatly valuable and significant for revising and improving this manuscript. 

# Reviewer 2

1.The authors have poorly responded to some of the comments including the number of mice subjects which is one of the most important comments.

Response: I am very sorry that I did not provide good feedback in response to the last comment. Insufficient number of mice is one of the limitations of this study and we will mention this limitation in the discussion section on lines 326-328. However, as stated in the last reply letter, the number of mice used in this study was formulated based on the literature of excellent journals in the field of this study, e.g. Allery, Frontiers in immunology, therefore, the results of this study could provide support for the main objective of this study. In addition, this animal model is stable and reliable, therefore, a small number of mice were used for the experiments in this study to meet animal welfare requirements. The reason why this animal model is stable and reliable are that the animal sensitization model and specific immunotherapy were designed, as previously described [1-3]. In addition, the results of this study can also support this view.

2. Furthermore, the authors should justify their selection of female over male mice otherwise the experiment should use both male and female mice to exclude the gender effect. There should also be replication of the animal experiment to ensure fidelity.

Response: Numerous studies indicate that female BALB/c mice are more susceptible to the development of allergic inflammation than male mice. For example, allergens-driven accumulation of neutrophils, eosinophils and macrophages were significantly higher in females compared to males [4]. group 2 innate lymphoid cells (ILC2s)-activating cytokines including IL-33, IL-7 and TSLP were more highly expressed in whole lung homogenate samples prepared from naive post pubertal female mouse lung than male mouse lung [5]. Female compared with male mice generally developed more pronounced antibody and inflammatory responses [6-9]. However, testosterone, signaling through the androgen receptor (AR), decreased Th2-mediated allergic inflammation and type 2 innate immune responses during allergic inflammation [10]. And androgen signaling suppresses cytokine production of Th2 cells by inducing DUSP-2 [11].Therefore, female BALB/c mice were used in this study. The use of female mice in this study has been mentioned in the title and abstract on lines 3, 23, and the address this limitation has been noted in the discussion section on lines 328-329 , as well as noting that the results may be different for male mice on lines 329-330 and stating that gender effects will be further investigated in future studies on lines 330-331.

Then, the experiment was not independently repeated is one of the limitations of this study, future research will remedy this limitation.

 1. Liu J, Yin J. Immunotherapy With Recombinant Alt a 1 Suppresses Allergic Asthma and Influences T Follicular Cells and Regulatory B Cells in Mice. Front Immunol. 2021;12:747730. https://doi.org/10.3389/fimmu.2021.747730 PMID: 34804031.

 2. Zhang Q, Xi G, Yin J. Artemisia sieversiana pollen allergy and immunotherapy in mice. Am J Transl Res. 2021;13(12):13654-13664. PMID: 35035704.

 3. Hesse L, Petersen AH, Nawijn MC. Methods for Experimental Allergen Immunotherapy: Subcutaneous and Sublingual Desensitization in Mouse Models of Allergic Asthma. Methods Mol Biol. 2021;2223:295-335. https://doi.org/10.1007/978-1-0716-1001-5_20 PMID: 33226602.

 4. Mostafa D, Hemshekhar M, Piyadasa H, Altieri A, Halayko AJ, Pascoe CD, et al. Characterization of sex-related differences in allergen house dust mite-challenged airway inflammation, in two different strains of mice. Sci Rep. 2022;12(1):20837. https://doi.org/10.1038/s41598-022-25327-7 PMID: 36460835.

 5. Matha L, Shim H, Steer CA, Yin YH, Martinez-Gonzalez I, Takei F. Female and male mouse lung group 2 innate lymphoid cells differ in gene expression profiles and cytokine production. PLoS One. 2019;14(3):e0214286. https://doi.org/10.1371/journal.pone.0214286 PMID: 30913260.

 6. Alberg T, Hansen JS, Lovik M, Nygaard UC. Particles influence allergic responses in mice--role of gender and particle size. J Toxicol Environ Health A. 2014;77(5):281-92. https://doi.org/10.1080/15287394.2013.863746 PMID: 24588227.

 7. Hansen JS, Alberg T, Rasmussen H, Lovik M, Nygaard UC. Determinants of experimental allergic responses: interactions between allergen dose, sex and age. Scand J Immunol. 2011;73(6):554-67. https://doi.org/10.1111/j.1365-3083.2011.02529.x PMID: 21323693.

 8. Okuyama K, Wada K, Chihara J, Takayanagi M, Ohno I. Sex-related splenocyte function in a murine model of allergic asthma. Clin Exp Allergy. 2008;38(7):1212-9. https://doi.org/10.1111/j.1365-2222.2008.03015.x PMID: 18498415.

 9. Melgert BN, Postma DS, Kuipers I, Geerlings M, Luinge MA, van der Strate BW, et al. Female mice are more susceptible to the development of allergic airway inflammation than male mice. Clin Exp Allergy. 2005;35(11):1496-503. https://doi.org/10.1111/j.1365-2222.2005.02362.x PMID: 16297148.

10. Fuseini H, Yung JA, Cephus JY, Zhang J, Goleniewska K, Polosukhin VV, et al. Testosterone Decreases House Dust Mite-Induced Type 2 and IL-17A-Mediated Airway Inflammation. J Immunol. 2018;201(7):1843-1854. https://doi.org/10.4049/jimmunol.1800293 PMID: 30127088.

11. Ejima A, Abe S, Shimba A, Sato S, Uehata T, Tani-Ichi S, et al. Androgens Alleviate Allergic Airway Inflammation by Suppressing Cytokine Production in Th2 Cells. J Immunol. 2022;209(6):1083-1094. https://doi.org/10.4049/jimmunol.2200294 PMID: 35977797.

---

## [Editor Report · Decision Letter 2]

11 Oct 2023

PONE-D-23-00019R2Recombinant Art v4.01 protein produces immunological tolerance by subcutaneous immunotherapy in a wormwood pollen-driven allergic asthma female mouse model .PLOS ONE

Dear Dr. Li,

Thank you for submitting your manuscript to PLOS ONE. After careful consideration, we feel that it has merit but does not fully meet PLOS ONE’s publication criteria as it currently stands. Therefore, we invite you to submit a revised version of the manuscript that addresses the points raised during the review process.

The investigator declined repeating the experiment to ensure rigor and reproducibility.

The explanation that the model is sufficiently robust is not convincing to me, because while model itself can be robust, the response to treatment with immunotherapy may vary.

Since the conclusion about the effectiveness of immunotherapy is firm and in the title, we requested the repeat of the most important readouts to ensure that the therapy works consistently.

The Animal committees issue the permission for additional mice when reviewers of the paper request it.

Therefore I insist that the repeat is conducted. The data can be pulled if they are similar in both repeats to increase the N values and power of the statistic or shown in the supplement.

We look forward to receiving your revised manuscript.

Kind regards,

Michal A Olszewski, DVM, PhD

Academic Editor

PLOS ONE

---

## [Author Response · Author response to Decision Letter 2]

15 Jan 2024

Dear Editor

Thank you for comments concerning our manuscript entitled “Recombinant Art v4.01 protein produces immunological tolerance by subcutaneous immunotherapy in a wormwood pollen-driven allergic asthma female mouse model”. Those comments were greatly valuable and significant for revising and improving this manuscript. In order to ensure rigor and reproducibility of this experiment, we repeated the most important readouts to ensure that the therapy works consistently. The data of the repeated experiment showed the same trend with the results of the previous experiment, indicating that the results of this experiment are credible and repeatable. Thus, the new data were shown in the supplement material. All the changes have been highlighted in red in the revised version of the manuscript, and we hope that the revised version of the manuscript would be acceptable for publication in your prestigious journal. The new data have been shown as follows:

Compared with the model group, the number of WBCs, TNCs, MNs, and PMNs in the BALF in the treatment group showed a downward trend, in which the number of WBCs, MNs in the rArt v4.01 group significantly decreased (P < 0.05, Fig. 1A), and the number of inflammatory cells in the extract group did not significantly vary (P > 0.05, Fig. 1A). The inflammation scores in the rArt v4.01 and extract groups [2.30±0.64 and 2.95±0.61, respectively] were significantly lower than those in the model group (4.20±1.19, P < 0.05, Fig. 1B). This part of the result was similar to the previous experimental results. It was demonstrated that both strategies mitigated allergic inflammation, including a decrease in the number of inflammatory cells in BALF, and reduced congestion and inflammatory cell infiltration in the alveolar wall. However, the ability of rArt v4.01 to inhibit inflammation was superior to that of wormwood.

In addition, serum IgE level in the rArt v4.01 group was lower than that in the model group (1.17±0.18) μg/ml (P = 0.051), while the extract group had no significant effect (P > 0.05, Fig. 2A). In contrast, serum IgG1 levels in the extract groups were 1.97±0.19 mg/ml, which were significantly higher than those in the model group (1.56±0.03 mg/ml, P < 0.05, Fig. 2A). Similar to IgG1, serum IgG2a level was upregulated in the SCIT group. Importantly, serum IgG2a level was significantly higher in the extract group than that in the model group (6.44±1.55 vs. 3.68±0.68 µg/ml) (P < 0.05), while rArt v4.01 had no significant effect (P > 0.05, Fig. 2A). This part of the result was also similar to the previous experimental results. Only recombinant protein treatment significantly reduced serum IgE level. And only the wormwood was effective in inducing serum levels of IgG1 and IgG2a, while rArt v4.01 did not produce similar results. 

The levels of Th2-related cytokines (IL-5 and IL-2) and Th17-related cytokines(IL-17A) were significantly downregulated in the rArt v4.01 and extract group compared with those in the model group (P < 0.05, Fig. 2B), with a greater decrease in the rArt v4.01 group compared to the extract group. TNF-α was significantly downregulated in the rArt v4.01 group compared with the model group (P < 0.05, Fig. 2B). In contrast, the levels of Treg-related cytokine (IL-10) and Th1-type cytokine (IFN-γ) were significantly upregulated in the extract group (P < 0.05), TGF-β significantly upregulated in the rArt v4.01 and extract group compared with those in the model group (P < 0.05, Fig. 2B). In addition, the levels of IL-4 did not significantly differ between the two treatment groups (P > 0.05, Fig. 2B). This part of the result was also similar to the previous experimental results. It was also confirmed that rArt v4.01 inhibited Th2 and Th17 response more strongly than wormwood, while induced Th1, regulatory T cells (Tregs) weaker than crude extract.

---

## [Editor Report · Decision Letter 3]

1 Feb 2024

PONE-D-23-00019R3Recombinant Art v4.01 protein produces immunological tolerance by subcutaneous immunotherapy in a wormwood pollen-driven allergic asthma female mouse model .PLOS ONE

Dear Dr. Li,

Thank you for submitting your manuscript to PLOS ONE. After careful consideration, we feel that it has merit but does not fully meet PLOS ONE’s publication criteria as it currently stands. Therefore, we invite you to submit a revised version of the manuscript that addresses the points raised during the review process.

The manuscript has significantly improved and the addition of new data is greatly appreciated. However, the new supplementary data have not been cited in the manuscript. Whenever you report the observation it would be good to refer to the supplementary figure as well. The authors may also include following note in the figure legend. Representative experiment of 2, repeat is shown in Fig S3. The independent repeats should be mentioned in the method section as well.  Finally, the marked up manuscript did not visualize changes as described in the letter.==============================

We look forward to receiving your revised manuscript.

Kind regards,

Michal A Olszewski, DVM, PhD

Academic Editor

PLOS ONE
---

## [Author Response · Author response to Decision Letter 3]

10 Mar 2024

Dear Editor

Thank you for comments concerning our manuscript entitled “Recombinant Art v4.01 protein produces immunological tolerance by subcutaneous immunotherapy in a wormwood pollen-driven allergic asthma female mouse model”. Those comments were greatly valuable and significant for revising and improving this manuscript. 

Firstly, the new supplementary data have been cited in the manuscript on lines 251-254, 256-258, 268-273, 278-286, 318-322 and324-328. Secondly, we include following note in the figure legend. Replicates of these experiments are shown in Fig S4, 5 (lines 160 and 241). Thirdly, The independent repeats have been mentioned in the method section on lines 202-204. Finally, All the changes have been highlighted in red in the revised version of the manuscript, and we hope that the revised version of the manuscript would be acceptable for publication in your prestigious journal.

---

## [Editor Report · Decision Letter 4]

11 Mar 2024

Recombinant Art v4.01 protein produces immunological tolerance by subcutaneous immunotherapy in a wormwood pollen-driven allergic asthma female mouse model .

PONE-D-23-00019R4

Dear Dr. Li,

We’re pleased to inform you that your manuscript has been judged scientifically suitable for publication and will be formally accepted for publication once it meets all outstanding technical requirements.

Kind regards,

Michal A Olszewski, DVM, PhD

Academic Editor

PLOS ONE

Additional Editor Comments (optional):

Congratulations on your revised manuscript and the final edits.
---

## [Editor Report · Acceptance letter]

14 May 2024

PONE-D-23-00019R4 

PLOS ONE

Dear Dr. Li, 

I'm pleased to inform you that your manuscript has been deemed suitable for publication in PLOS ONE. Congratulations! Your manuscript is now being handed over to our production team.

Kind regards, 

on behalf of

Dr. Michal A Olszewski 

Academic Editor

PLOS ONE